# Effective use of a horizontally-transferred pathway for dichloromethane catabolism requires post–transfer refinement

Joshua K Michener[1], Aline A Camargo Neves[1,2], Stéphane Vuilleumier[3], Françoise Bringel[3], Christopher J Marx[1,4,5,6]*

[1]Department of Organismic and Evolutionary Biology, Harvard University, Cambridge, United States; [2]Department of Microbiology, Institute of Biomedical Sciences, University of São Paulo, São Paulo, Brazil; [3]CNRS Molecular Genetics, Genomics, Microbiology, Université de Strasbourg, Strasbourg, France; [4]Faculty of Arts and Sciences Center for Systems Biology, Harvard University, Cambridge, United States; [5]Department of Biological Sciences, University of Idaho, Moscow, United States; [6]Institute for Bioinformatics and Evolutionary Studies, University of Idaho, Moscow, United States

**Abstract** When microbes acquire new abilities through horizontal gene transfer, the genes and pathways must function under conditions with which they did not coevolve. If newly-acquired genes burden the host, their utility will depend on further evolutionary refinement of the recombinant strain. We used laboratory evolution to recapitulate this process of transfer and refinement, demonstrating that effective use of an introduced dichloromethane degradation pathway required one of several mutations to the bacterial host that are predicted to increase chloride efflux. We then used this knowledge to identify parallel, beneficial mutations that independently evolved in two natural dichloromethane-degrading strains. Finally, we constructed a synthetic mobile genetic element carrying both the degradation pathway and a chloride exporter, which preempted the adaptive process and directly enabled effective dichloromethane degradation across diverse *Methylobacterium* environmental isolates. Our results demonstrate the importance of post–transfer refinement in horizontal gene transfer, with potential applications in bioremediation and synthetic biology.

*For correspondence: cmarx@uidaho.edu

**Reviewing editor**: Michael Laub, Massachusetts Institute of Technology, United States

## Introduction

Microbes frequently acquire genetic material from organisms that are distant evolutionary relatives but close ecological neighbors (*Ochman et al., 2000*; *Gogarten et al., 2002*; *Smillie et al., 2011*). These horizontally-transferred genes and pathways play important roles in processes ranging from the spread of antibiotic resistance to ecological differentiation (*Forsberg et al., 2012*; *Shapiro et al., 2012*). When, for example, a new environmental niche opens up through the introduction of a xenobiotic compound, horizontal gene transfer (HGT) can speed the assembly and dissemination of a corresponding catabolic pathway (*Springael and Top, 2004*). However, newly-acquired abilities may be costly for the host until they are carefully integrated into existing metabolic and regulatory networks (*Kim and Copley, 2012*; *Yadid et al., 2013*). The challenges associated with such adaptations are highlighted by many examples from metabolic engineering, where productively transferring genes and pathways into new hosts often requires significant post–transfer refinement (*Chang et al., 2007*; *Michener et al., 2012*).

Observing this process of transfer and refinement in nature is challenging, but we can recreate the same process in the laboratory using a combination of genetic engineering and experimental

**eLife digest** Many microbes can rapidly evolve to adapt to new or extreme habitats. Most often the characteristics that develop via evolution result from individuals inheriting new combinations of genes from their parents. However, many species can also acquire new genes through a process called horizontal gene transfer, where organisms that share an environment exchange potentially useful genes. The ability of bacteria to acquire genes by horizontal gene transfer is thought to be the main reason for the rapid evolution of antibiotic-resistant bacteria such as MRSA.

A horizontally transferred gene rarely works efficiently when first transferred, and may even cause problems for its new host. The host organism must therefore evolve further after receiving a new gene, but it is difficult to trace how this occurs.

A toxic chemical called dichloromethane (or DCM) has been used in many industries since World War II, and has caused widespread contamination to the environment. Strains from several microbial genera have been identified that have adapted to break down DCM and use it as their sole energy source. Moreover, a gene responsible for breaking down DCM appears to have been shared between different species by horizontal gene transfer.

In work that was presented earlier in 2014, researchers introduced this gene into several bacterial strains from the genus *Methylobacterium* that had not been previously exposed to DCM. However, the resulting new strains of bacteria still had difficulties growing on DCM. There were diverse ways that the bacteria could have been prevented from growing—for example, multiple by-products generated when DCM is broken down are highly toxic.

Now Michener et al.—including the researchers that performed the 2014 work—have experimentally evolved five of the modified strains of *Methylobacterium* to discover how they adapt over time to living on DCM. Examining the DNA sequences of these bacteria showed several mutations that both improved the ability of the bacteria to survive on DCM and helped the bacteria to more efficiently remove chloride ions from their systems. Chloride ions accumulate as DCM is broken down; therefore, Michener et al. suggest that being exposed to too much chloride prevented the bacteria in previous experiments from growing well. This idea is further supported by mutations found in previously isolated bacteria that live on DCM. These mutations increase the bacteria's ability to excrete chloride ions from their cells, and when Michener et al. removed these mutations, the bacteria no longer grew well on DCM.

Understanding how bacteria adapt once they have acquired the DCM-degrading gene could make it easier to help bacteria to mop up DCM and other contaminants in the environment.

evolution. Laboratory evolution has been used extensively to study natural evolutionary processes. Recent experiments have highlighted the mechanisms by which evolution can optimize existing traits (*Elena and Lenski, 2003*), select for the emergence of a novel ability (*Blount et al., 2012*), and refine a rudimentary pathway (*Quandt et al., 2014*). Laboratory studies focusing on genetic exchanges between microbes have described the recombination of mutations within an evolving population (*Winkler and Kao, 2012*) or the replacement of an endogenous metabolic pathway with an alternative route (*Chou et al., 2011*). Meanwhile, metabolic engineers have used adaptive evolution as an engineering tool, including the evolution of highly modified strains, but under conditions that do not reproduce HGT (*Fong et al., 2005*; *Trinh and Srienc, 2009*; *Lee and Palsson, 2010*; *Lee et al., 2012*). In this work, we combined these approaches to study how microbes evolve following the acquisition through horizontal gene transfer of a novel metabolic ability by deliberately transferring a mobile metabolic pathway into a new host and then using experimental evolution to select for its efficient use.

We investigated this process of HGT followed by evolutionary refinement using dichloromethane (DCM) catabolism in *Methylobacterium* strains as an example. Dichloromethane became an important industrial solvent during World War II and is still the most prevalent chlorinated solvent today (www.eurochlor.org). Following widespread environmental contamination by DCM, several microbial strains have been isolated based on their ability to grow on DCM as the sole source of carbon and energy. Most notably, *Methylobacterium extorquens* DM4, a methylotrophic Alphaproteobacterium,

has served as a reference model to elucidate the details of bacterial growth with DCM (*Gälli and Leisinger, 1985*; *Muller et al., 2011a*). *M. extorquens* DM4 expresses a dehalogenase, DcmA, whose gene shows clear signs of horizontal transfer (*Schmid-Appert et al., 1997*; *Vuilleumier et al., 2009*) and whose product, formaldehyde, feeds directly into central methylotrophic metabolism (*Chistoserdova, 2011*).

Past efforts to understand the genetics of growth on DCM used gene knockouts to uncover DCM-specific genes (*Figure 1A*) (*Muller et al., 2011b*). We complemented these efforts using a synthetic approach involving conjugal transfer of the *dcmA* gene into several naïve *Methylobacterium* strains (*Figure 1B*) (*Kayser et al., 2000*; *Michener et al., 2014*). These *dcmA* transconjugants showed DCM dehalogenase activity but grew poorly on DCM, and we previously investigated factors that might explain their limited growth (*Michener et al., 2014*). Growing on DCM using the *dcmA* catabolic pathway is very challenging for the host, as its products and intermediates include formaldehyde, hydrochloric acid, and the alkylating agent *S*-chloromethylglutathione (*Figure 1F*) (*Kayser et al., 2000*; *Kayser and Vuilleumier, 2001*). As DCM dehalogenase activity was generally high in transconjugant strains (*Michener et al., 2014*), we hypothesized that one or more of these stresses was limiting growth in the transconjugants and thus required post–transfer evolutionary refinement of the host to produce highly active DCM-degrading strains.

In this work, we used experimental evolution to identify the factors that were limiting growth in *dcmA*-containing transconjugants. Beginning from these ancestral transconjugants, we evolved replicate populations with DCM as the sole source of carbon and energy and obtained evolved isolates with substantially higher fitness on DCM (*Figure 1C*). We sequenced the genomes of these evolved strains to identify the mutations that increased fitness on DCM and then reconstructed these mutations in isogenic backgrounds to measure their effect on fitness (*Figure 1D*). Based on these mutations, we could infer both the primary limiting stress that the pathway placed on its host as well as the biochemical mechanisms that the cells used to overcome this stress. Finally, we used this knowledge to uncover naturally-occurring refining mutations in two DCM-degrading environmental isolates and to design an improved gene cassette that enhances DCM bioremediation by preempting the need for post–transfer refinement (*Figure 1E*).

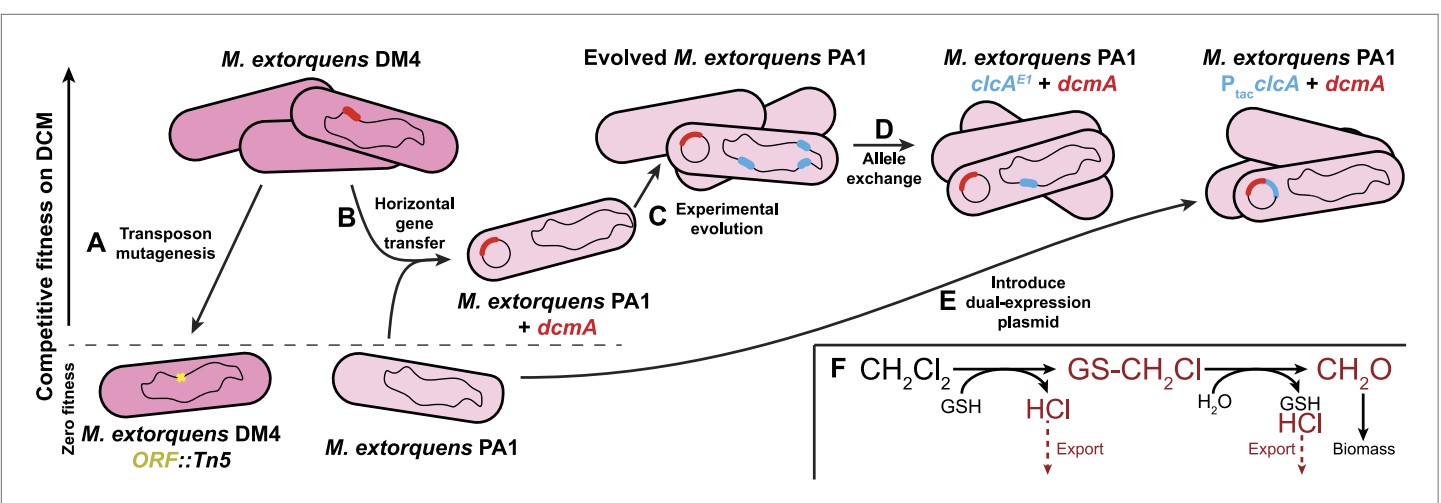

**Figure 1**. Experimental evolution recapitulates post–transfer optimization of a challenging catabolic pathway. The natural isolate, DM4, grows well on dichloromethane. (**A**) Past efforts to explain the genetics of growth on DCM relied on gene knockouts (yellow) to identify important genes (*Muller et al., 2011b*). (**B**) We deliberately transferred the *dcmA* gene (red) into naïve recipient *Methylobacterium* strains, which then grew poorly on DCM (*Michener et al., 2014*). (**C**) Serial propagation on DCM selected for mutants with increased fitness on DCM. (**D**) Whole genome resequencing allowed us to identify these mutations (blue), and reconstructing the individual mutations in wild type cells verified that they were causal. We then worked backwards from the mutations to identify the stress that the mutations overcame. (**E**) Introducing a plasmid containing both the pathway (red) and a solution to the most common limiting stress (blue) allowed efficient growth on DCM without chromosomal modifications to the host. (**F**) The biochemistry of DCM dehalogenation produces several challenging compounds (red) that potentially limit growth on DCM.

## Results

### Experimental evolution selected for increased fitness on DCM

We previously transferred the DCM dehalogenase *dcmA* into three strains of *M. extorquens* as well as three other species of *Methylobacterium* (*Michener et al., 2014*). Each of the transconjugants was significantly less fit than the natural isolate, *M. extorquens* DM4 (hereafter referred to as 'DM4'). In this work, replicate populations of five of these transconjugants were serially propagated with DCM as the sole carbon and energy source to select for increased fitness on DCM. One of these five transconjugants, *M. extorquens* AM1, was initially unable to grow on DCM alone and instead was propagated on a mixture of DCM and methanol. After a total of 150 generations of growth, individual clones were isolated from each replicate population. We measured the fitness of each clone in direct competition with the reference strain, DM4, and selected the most-fit clone from each population for further analysis.

For four out of the five ancestral transconjugant strains, the fitness of isolates from each evolved population was significantly higher than the fitness of the ancestor during growth on DCM (*Figure 2*), increasing by 1.6- to 13-fold relative to the respective ancestral transconjugant. *Methylobacterium radiotolerans* was the only transconjugant that did not yield improved isolates after 150 generations, and these non-improved populations were not further characterized.

### Whole-genome resequencing and allele exchange identified causal mutations

The genomes of all ancestral strains had previously been sequenced (*Vuilleumier et al., 2009*; *Marx et al., 2012*), so we used a combination of whole-genome resequencing and targeted Sanger sequencing to identify putative causal mutations in the evolved isolates. In the seven isolates that we resequenced, we repeatedly found mutations to the genes for the protein translocase *secY*, the chloride/proton antiporter *clcA*, and a hypothetical protein with a domain of unknown function (DUF599, which we renamed *edgA* for Evolved Dichloromethane Growth), as well as a single mutation to *besA*, a homolog of the eukaryal bestrophin family of chloride channels (*Figure 3A*).

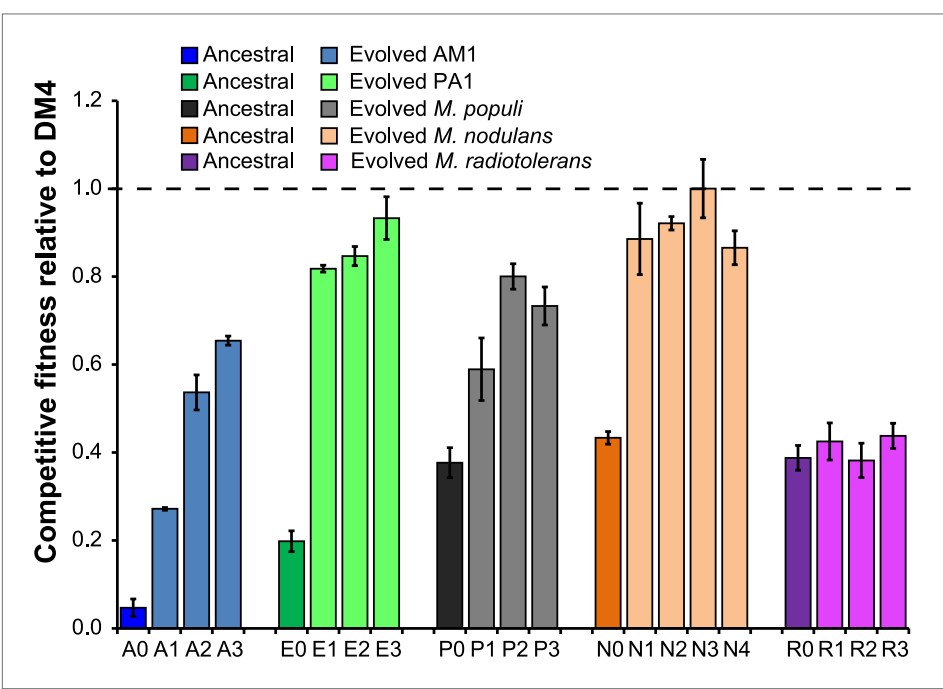

**Figure 2**. Evolved isolates from four of the five transconjugants have improved fitness relative to the ancestor. A single clonal isolate was selected from each replicate population after 150 generations of growth on DCM. Each isolate was mixed with the reference strain, DM4Δ*dcmA*+pJM10, and grown on DCM to determine competitive fitness. Error bars show ±1 standard deviation, calculated from three biological replicates. The horizontal dashed line indicates equal fitness to the reference isolate, DM4.

Sanger sequencing confirmed these mutations in the resequenced isolates and targeted sequencing of these loci demonstrated that each remaining isolate had mutations at one of these four loci. We note that additional mutations were identified in each resequenced isolate (*Figure 3—source data 1*).

Repeated mutations to the same loci in independently evolved replicates strongly suggested that the mutations were causal. However, to conclusively demonstrate causality we introduced several of these mutations into the ancestral strains *M. extorquens* AM1 and PA1 (hereafter referred to as 'AM1' and 'PA1'). We focused here upon AM1 and PA1, as these two strains incurred mutations to all four loci described above and are genetically tractable. Since the *secY* mutation was not identified in PA1 nor the *clcA* mutation in AM1, we also constructed hybrid allele exchange vectors to move these mutations into the hosts in which they were not observed. We then measured the fitness of the reconstructed mutants against the natural isolate, DM4. Each of the mutations was highly beneficial, regardless of whether the mutation arose in that host during evolution (*Figure 3B*). In contrast, the mutations had only minor fitness effects during growth on an alternate carbon source, succinate, indicating that their beneficial effects were specific to the challenge of growing on DCM and were not broadly beneficial during laboratory growth (*Figure 3—figure supplement 1*).

While each of the single mutations was beneficial, the fitness effects were not additive. For example, isolate E2 contained mutations in both *clcA* and *besA*. Each of these mutations was beneficial

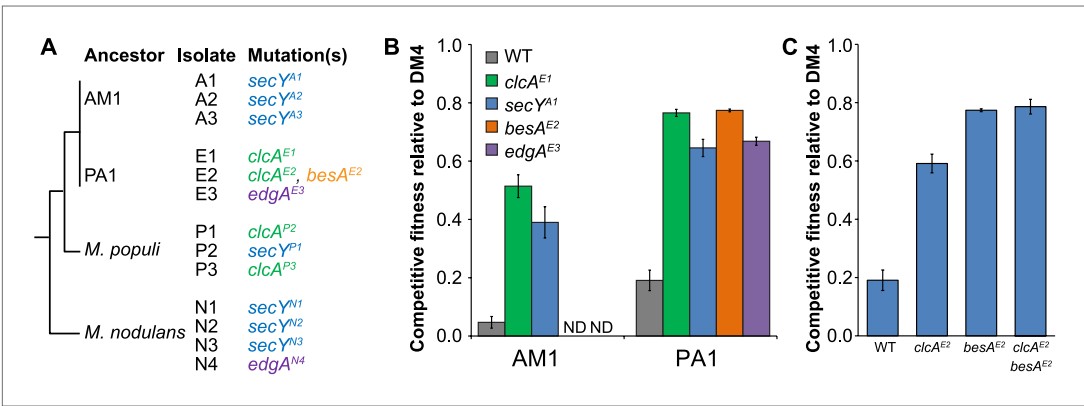

**Figure 3**. Each evolved isolate has a mutation in at least one of four loci. (**A**) A combination of whole genome resequencing and targeted Sanger sequencing allowed the identification of mutations at four loci, encoding the protein translocase *secY*, the chloride/proton antiporter *clcA*, a hypothetical protein that we named *edgA*, and the bestrophin-homolog chloride channel *besA*. A simplified phylogenetic tree is shown at left (*Michener et al., 2014*). Mutation names indicate the genetic locus followed by the isolate name in which it has been detected. (**B**) Reconstruction of these single mutations in AM1 and PA1 demonstrated that each individual mutation was beneficial, even in a host in which the mutation was not observed (*secY* in PA1 and *clcA* in AM1). (**C**) In PA1, the double mutant *besA*E2/*clcA*E2 has similar fitness to the *besA*E2 single mutant when grown on DCM. During the evolution of population E2, the *clcA*E2 mutation fixed first, followed by the *besA*E2 mutation. Each of the reconstructed strains expresses *dcmA* from plasmid pJM10. Error bars show ±1 standard deviation, calculated from three biological replicates. N.D.: not determined.

The following source data and figure supplements are available for figure 3:

**Source data 1**. Mutations identified during experimental evolution. N.D.: not determined.

**Figure supplement 1**. Mutations have relatively small fitness effects during growth on succinate compared to the benefits of threefold or more on DCM.

**Figure supplement 2**. Comparison of competitive fitness of single and double mutants in *M. extorquens* PA1 explains evolutionary patterns.

**Figure supplement 3**. Allele frequency dynamics for *M. nodulans* populations.

**Figure supplement 4**. In population E2, the *clcA*E2 mutation fixed first, followed by *besA*E2.

alone, but the double mutant was no more fit than the *besA*[E2] single mutant (*Figure 3C*). Similarly, when we constructed *clcA*[E1]/*secY*[A1] and *clcA*[E1]/*edgA*[E3] double mutants, we found that the double mutants were less fit than the *clcA*[E1] single mutant (*Figure 3—figure supplement 2*).

## Mutations to secY and clcA are predicted to increase chloride export

The protein translocase gene *secY* was mutated in seven of thirteen evolved isolates, totaling six amino acid mutations and one in-frame deletion. In the crystal structure of the SecY homolog from *Methanocaldococcus jannaschii* (*Van den Berg et al., 2004*), these seven mutations cluster to the same region of the protein surface, at the interface of the channel pore and its blocking plug (*Figure 4A*). Several of these mutations have been identified and characterized in the *Escherichia coli* SecY homolog (with 58% amino acid identity to *M. extorquens* DM4 SecY) based on their ability to secrete proteins with mistranslated sequences (*Emr et al., 1981*). These mutations are predicted to disrupt interactions between the plug and pore, leading to a leaky channel (*Smith et al., 2005*), and detailed biochemical investigation of one such mutant in *E. coli* demonstrated

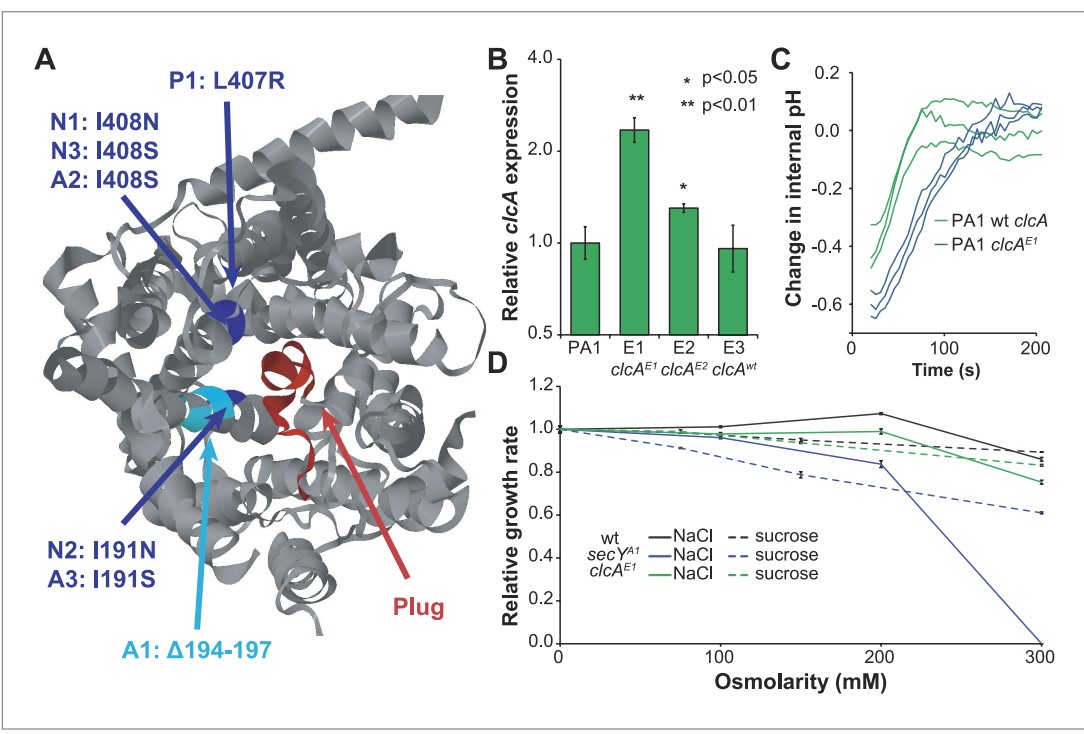

**Figure 4**. Mutations in *secY* and *clcA* are predicted to increase chloride export. (**A**) When mapped to the crystal structure of SecY from *Methanocaldococcus jannaschii* (*Van den Berg et al., 2004*), the seven mutations to *secY* cluster in a small region of the protein structure (blue and cyan) at the interface between the channel and plug (red) that is associated with leakage of small anions. (**B**) Mutations to the *clcA* promoter in isolates E1 and E2 lead to significantly increased transcription of *clcA*, as measured by qRT-PCR (one tailed Welch's t-test with two degrees of freedom). Error bars show ±1 standard deviation, calculated from three biological replicates. (**C**) Adding DCM to strains that overexpress *clcA* leads to a larger decrease in internal pH. Strains contain pJM40, expressing both *dcmA* and a fluorescent pH biosensor. The strain with a mutated *clcA* promoter (blue) overexpresses ClcA relative to the wild type strain (green). At t = 0, DCM was added to a final concentration of 10 mM. After an approximate delay of 20 s, the culture was continuously sampled with a flow cytometer to determine the dynamics of the internal pH. Both strains showed a transient decrease in pH, but the decrease is larger for the *clcA* overexpression strain (p = 0.036, one-tailed Welch's t-test with two degrees of freedom). (**D**) Mutations to *secY*, but not to *clcA*, increase chloride sensitivity. Growth rates of three PA1 strains (wild type, *secY*[A1], and *clcA*[E1]) were measured using an automated assay system. The osmolarity of the medium was varied through the addition of sucrose or sodium chloride. The *secY*[A1] mutant did not grow at the highest chloride concentration.

The following figure supplement is available for figure 4:

**Figure supplement 1**. Relative *clcA* expression is independent of the growth substrate.

that this mutation allows the facilitated diffusion of small anions, most notably chloride (*Dalal and Duong, 2009*).

Similarly, the promoter of the chloride/proton antiporter *clcA* was mutated in four of the thirteen evolved isolates. The two evolved PA1 isolates with *clcA* promoter mutations showed increased levels of *clcA* mRNA compared to either the ancestor or the PA1 isolate with a wild type *clcA* promoter (*Figure 4B* and *Figure 4—figure supplement 1*). However, the *E. coli* ClcA antiporter (with 33% amino acid identity to DM4) is capable of importing or exporting chloride, while necessarily transporting protons in the opposite direction (*Accardi and Miller, 2004*). To determine the directionality of ClcA-mediated transport in our strains, we used a pH biosensor to determine the effect of *clcA* overexpression on proton transport during growth with DCM. We observed a transient decrease in intracellular pH upon addition of DCM that was significantly larger in the strain overexpressing *clcA* (*Figure 4C*), consistent with ClcA importing protons and exporting chloride.

Next, we tested the effect of *secY* and *clcA* mutations on the chloride sensitivity of the mutant strains, since mutations that benefit the cell by facilitating chloride diffusion during growth on DCM may also result in detrimental levels of chloride import in high salt media. We measured the growth rate of strains on succinate with varying external chloride concentrations, using sucrose as a control for the effects of osmolarity. We were unable to detect a change in the sensitivity of AM1 mutants because the strain was already highly chloride sensitive (*Michener et al., 2014*). In PA1, the *clcA*$^{E1}$ mutant was no more sensitive to NaCl or sucrose than the wild type strain (*Figure 4D*). In contrast, the PA1 *secY*$^{A1}$ mutant was highly sensitive to NaCl, but only slightly more sensitive to sucrose (*Figure 4D*).

## Two DCM-degrading environmental *Methylobacterium* isolates contain mutations to *clcA*

We next asked whether our laboratory evolution experiments had recapitulated the natural process of adaptation to growth on DCM, thereby allowing the identification of refining mutations in environmental isolates. The sequences of *secY*, *besA*, and *edgA* in the reference isolate, DM4, did not contain any of the mutations identified in the laboratory-evolved strains. However, we found that DM4 expressed significantly higher levels of *clcA* mRNA than the other three ancestral strains of *M. extorquens* tested (*Figure 5A*). The sequences upstream of *clcA* differ between all four strains, making it difficult to identify the causal mutations in DM4. Consequently, we exchanged a 140-bp region upstream of *clcA* (containing a total of five nucleotide changes) between DM4 and PA1. The resulting DM4 strain with the PA1 *clcA* promoter had decreased *clcA* expression and minimal growth with DCM as the sole carbon and energy source, while the PA1 strain with the DM4 *clcA* promoter had increased *clcA* expression and was significantly more fit than the wild type PA1 strain (*Figure 5B*). However, complementing the DM4 *clcA*$^{E0}$ mutant strain with a *clcA*/*dcmA* expression plasmid restored growth on DCM (*Figure 6—figure supplement 1*).

Having identified necessary refining mutations in DM4, we examined the other known DCM-degrading *Methylobacterium* isolate, *M. extorquens* DM17 (*Firsova et al., 2010*), to determine whether a parallel event had occurred in its evolutionary history. When we sequenced the *secY*, *besA*, and *edgA* loci in this strain, the nucleotide sequences were exactly identical to *M. extorquens* AM1. Compared to *M. extorquens* AM1, however, the *clcA* locus of DM17 contained an insertion sequence 52 bp upstream of the *clcA* promoter, and we found that *clcA* expression was even higher in DM17 than in DM4 (*Figure 5A*).

## A synthetic expression cassette containing both *clcA* and *dcmA* confers higher fitness on DCM in naïve transconjugants

Having demonstrated that *clcA* overexpression increases the fitness in several strains of *M. extorquens* during growth on DCM, we tested whether this effect would also hold in other environmental isolates across the *Methylobacterium* genus. We constructed a plasmid that expressed *dcmA* from the native DM4 promoter as well as *clcA* from a constitutive promoter. We then mated either our original *dcmA* plasmid (pJM10) or this new *dcmA*/*clcA* plasmid (pJM83) into a collection of natural *Methylobacterium* isolates. We found that every strain showed higher fitness on DCM with the dual-expression *dcmA*/*clcA* plasmid than with the single-expression *dcmA* plasmid (*Figure 6*).

## Discussion

Previous studies in experimental evolution generally focused on a single wild type microbial strain. In contrast, we evolved five different *Methylobacterium* strains, each of which carried the same

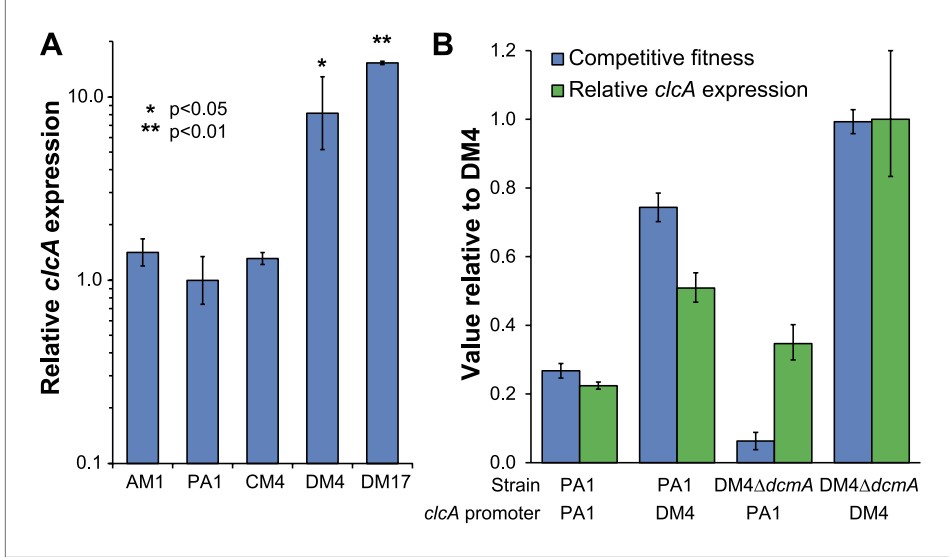

**Figure 5**. DM4 requires *clcA* overexpression to grow on DCM. (**A**) The environmental DCM-degrading isolates, DM4 and DM17, overexpress *clcA* relative to three other strains of *M. extorquens* (one tailed Welch's t-test with two degrees of freedom). Error bars show ±1 standard deviation, calculated from three biological replicates. (**B**) Growth on DCM in DM4 and PA1 depends on the *clcA* promoter but not the strain background. A 140-bp region upstream of *clcA* (the '*clcA* promoter') was swapped between PA1 and DM4. Regardless of the genetic background, strains with the DM4 *clcA* promoter overexpressed *clcA* and were more fit on DCM compared to strains with the PA1 *clcA* promoter. All strains expressed *dcmA* from plasmid pJM10. Error bars show ±1 standard deviation, calculated from three biological replicates.

horizontally-transferred metabolic pathway for DCM catabolism. By sequencing the evolved isolates, we identified mutations across diverse backgrounds that improve the ability to grow on DCM. Working backward from these mutations allowed us to determine that chloride accumulation appears to be the most critical factor limiting growth, and to identify multiple mechanisms that alleviate this stress. We found parallel responses at both the genetic and physiological level, indicating that the different strains faced similar challenges when forced to use their new metabolic capability and overcame these challenges in similar ways. Horizontal transfer of *dcmA* is not sufficient for efficient growth on DCM and, in both the laboratory and the environment, microbes must also acquire refining mutations to optimize the newly-acquired pathway.

Several lines of evidence suggest that chloride export was the key limiting factor during growth on DCM. First, the observed mutations in *secY* are predicted to turn the protein translocase into a leaky chloride channel, as demonstrated previously in *E. coli* (*Dalal and Duong, 2009*). Our growth rate measurements confirm that *secY* mutants are more sensitive to extracellular chloride, as we would expect if the mutant SecY facilitates chloride diffusion across the cell membrane. Second, mutations to the promoter of *clcA* increase *clcA* expression. The in vivo pH measurements upon DCM addition establish that ClcA imports protons, with a necessary concomitant export of chloride, and that increased ClcA expression leads to increased proton import and chloride export. Similarly, the *E. coli* response to extreme acid stress uses ClcA to export chloride, with other mechanisms used for proton export (*Accardi and Miller, 2004*). Third, BesA is predicted to be a chloride channel (*Sun et al., 2002*) and while we cannot yet explain the mechanism by which the identified mutation would affect its function, we hypothesize that the mutation may also allow increased chloride export. Finally, we observed that double mutants are no more fit than the best single mutant, suggesting that all the mutations we detected are addressing the same physiological limitation. Previous experiments found that mutations affecting the same pathway were less beneficial when combined (*Chou et al., 2011*), in some cases to the point of being collectively deleterious (*Kvitek and Sherlock, 2011*; *Rokyta et al., 2011*; *Chou et al., 2014*). Due to the antagonism of *edgA*[E3] with *clcA*[E1], it is likely that EdgA is also involved in chloride export.

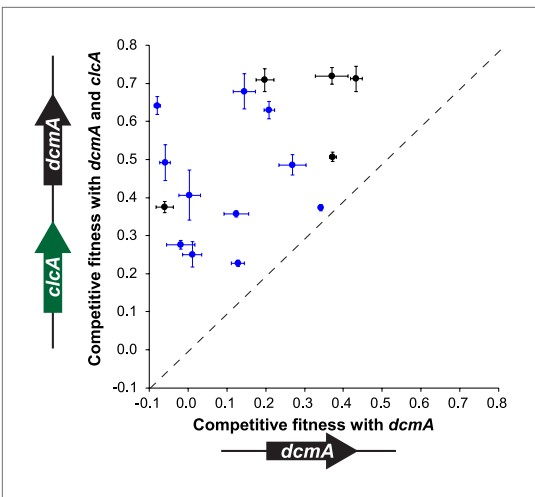

**Figure 6**. A mobile genetic element containing both *dcmA* and *clcA* allows a diverse array of *Methylobacterium* isolates to grow efficiently on DCM. Plasmids containing either *dcmA* (x-axis) or *dcmA* and *clcA* (y-axis) were introduced into 16 different *Methylobacterium* strains, including the five strains used to found the evolved populations (black) and 11 that were new (blue). Each transconjugant was tested against the reference, DM4Δ*dcmA*+pJM10, to determine its competitive fitness. Note that negative fitness values indicate net death over the course of the competition. The dashed line indicates equal fitness of the two transconjugants, as would be expected if *clcA* expression were neutral. Error bars show ±1 standard deviation, calculated from three biological replicates.

The following source data and figure supplement is available for figure 6:

**Source data 1**. Environmental *Methylobacterium* strains characterized with the dual expression plasmid.

**Figure supplement 1**. Overexpression of *clcA*, either by the DM4 *clcA* promoter or the pJM83 *clcA/dcmA* dual expression plasmid, confers high fitness on DCM.

While chloride export is not an unexpected stress during growth on DCM, there was no a priori reason to expect that chloride export was the major limiting stress across all of the strains that we tested. Multiple other stresses could have been dominant, including sensitivity to internal acidification, the solvent effects of DCM, changes in formaldehyde fluxes perturbing the native host metabolism, or the mutagenic effects of glutathione-conjugated DCM (*Kayser and Vuilleumier, 2001*). Indeed, our previous efforts to find a single physiological difference that explained the varied fitness of the ancestral *dcmA* transconjugants on DCM did not identify chloride sensitivity as a predictive phenotype for fitness during growth on DCM (*Michener et al., 2014*). There was no reason to expect that chloride was more toxic than protons, but using ClcA to export the chloride, even at the cost of importing yet more protons, was highly beneficial. It appears that excess intracellular chloride is either more toxic or less effectively managed than excess intracellular protons.

The apparent linkage between *besA* and chloride export may offer insights into a human disease. Mutations to human bestrophin homologs can lead to vitelliform macular dystrophy, but little is known about these proteins aside from their roles as chloride channels (*Sun et al., 2002*; *Tsunenari et al., 2003*). Bacterial homologs of the bestrophin family are similarly uncharacterized. Select bacterial homologs have been shown to affect swarming (*Inoue et al., 2007*) and sporulation (*Rahn-Lee et al., 2009*), but these systems have not explicitly replicated the role of the human homolog as a chloride channel. However, our *Methylobacterium* system may offer an opportunity to characterize the chloride transporter function of this protein family in a tractable host with a strong linkage between BesA function and fitness.

By design, our analysis has focused on mutations with very large fitness effects. Mutations that alleviate stresses imposed during growth on DCM had the potential to dramatically increase fitness, and restricting our evolution experiment to 150 generations predominantly selected these mutations. Our populations undoubtedly sampled mutations that would be generally beneficial during growth in our media and laboratory conditions, as has been seen in earlier experiments with *M. extorquens* AM1 (*Lee et al., 2009*), but the fitness effects of these mutations appear to have been minor compared to the DCM-specific mutations. For example, isolate N1 lost an entire 458 kbp plasmid, which was likely due to selection but with a small fitness benefit as compared to acquiring a *secY* mutation (*Lee and Marx, 2012*). The rarity of mutations generically beneficial during laboratory growth greatly simplified our analysis of the evolved isolates.

We expect that multiple lineages within a population independently arrived at solutions to increase chloride efflux. However, the size of the fitness effects produced evolutionary dynamics that resembled periodic selection (*Atwood et al., 1951*), in which the first large effect mutation to arise in the population rapidly fixed. Combinations of mutations were generally no more beneficial than the best single mutation, so we predominantly observed the result of a single selective

sweep (*Figure 3—figure supplement 3*). Population E2 is the exception, where the *clcA*[E2] mutation fixed first, followed by the *besA*[E2] mutation (*Figure 3—figure supplement 4*). The *clcA*[E2] mutant has a fitness of 0.6 as compared to 0.8 for *besA*[E2] or the *besA*[E2]/*clcA*[E2] double mutant (*Figure 3C*). Thus, while the double mutant is no more fit than the *besA*[E2] single mutant, the *besA*[E2] mutation is still highly beneficial in the *clcA*[E2] background.

Experimental populations of *M. radiotolerans* did not increase in fitness over 150 generations of growth on DCM. When we tested this strain with the dual-expression *clcA*/*dcmA* plasmid, the fitness increased but the increase was smaller than for other strains (*Figure 6—source data 1*). We hypothesize that our inability to find improved isolates of *M. radiotolerans* stems from a combination of fewer accessible beneficial mutations, smaller fitness effects of the beneficial mutations, and a decreased sensitivity to the mutagenic effects of growing on DCM (*Perez-Pantoja et al., 2013*).

Experimental evolution is increasingly used to simulate natural evolutionary processes (*Kawecki et al., 2012*); however, there are few examples where mutations that arise during laboratory evolution can also be identified in natural populations (*Wong et al., 2012*; *Traverse et al., 2013*). We have shown that mutations to *clcA* can be highly beneficial during laboratory growth, and that parallel mutations to *clcA* have arisen in two independent DCM-degrading environmental isolates of *M. extorquens*. In the reference environmental isolate *M. extorquens* DM4, we can directly link nucleotide mutations in the *clcA* promoter to *clcA* overexpression and increased organismal fitness during growth on DCM. Demonstrating linkages between evolutionary processes observed in the laboratory and in nature will be increasingly important for the further development and application of experimental evolution.

Having identified *clcA* overexpression as a beneficial evolutionary refinement following acquisition of *dcmA*, we used this knowledge to develop a novel bioremediation strategy and designed new expression cassettes for bioremediation that are less dependent on subsequent adaptation for effective function. Rather than introducing an exogenous microbe into a contaminated site, we propose to introduce a genetic cassette to the indigenous microflora, an approach known as genetic bioaugmentation (*Top et al., 2002*; *Ikuma and Gunsch, 2013*). Our experiments clearly demonstrated that simply introducing the degradation pathway for a contaminant can be inefficient, as the recipients may be unprepared for the stresses produced by the pathway (*Kayser et al., 2002*; *Michener et al., 2014*). Instead, we propose to identify these stresses in the laboratory and then provide both the catabolic pathway and solutions to the most common limitations. In our example of DCM degradation, transconjugants that received both the catabolic gene *dcmA* and the chloride exporter gene *clcA* were more efficient at degrading DCM than those that received *dcmA* alone (*Figure 6*).

In summary, our results highlight the important role of evolutionary refinement in the horizontal transfer of a challenging catabolic pathway. Dichloromethane degradation led to chloride accumulation and required modifications to the recipient to export the chloride. Based on recent examples from metabolic engineering, we expect that these types of host–pathway interactions are common (*Kizer et al., 2008*; *Ro et al., 2008*; *Verwaal et al., 2010*; *Michener et al., 2012*; *Bernhardt and Urlacher, 2014*) and, consequently, that post–transfer refinement of microbial hosts is also widespread in nature. Our approach of deliberate horizontal transfer in the laboratory followed by experimental evolution offers an opportunity to identify both the limiting stresses and the mutations that can overcome these stresses. As screens and selections for complex microbial phenotypes are further developed, we expect that this approach will find many applications, both in the analysis of natural microbial isolates and in the optimization of genetically engineered microbes and mobile genetic elements.

## Materials and methods

### Strains and cultivation

*E. coli* cultures were grown in LB containing 10 g/l NaCl except for cultures containing the *dcmA*/*clcA* dual expression plasmid (pJM83), which were salt-sensitive and instead were grown in LB containing 0.5 g/l NaCl. Unless otherwise noted, *Methylobacterium* strains were grown in 10 ml cultures in a 50 ml flask at 30°C and 220 rpm in M-PIPES medium supplemented with 3.5 mM succinate or 5 mM DCM (*Delaney et al., 2013a*). Antibiotics were added to final concentrations of 10 μg/ml streptomycin, 12.5 μg/ml tetracycline, or 50 μg/ml kanamycin. All DCM cultures were grown in gas-tight 50 ml screw-top flasks sealed with Mininert valves (Supelco, Bellefonte, PA) and teflon tape. Before use, valves were sterilized with ethanol and dried in a laminar flow hood. Unless otherwise noted, chemicals

were purchased from Sigma–Aldrich (St. Louis, MO) and enzymes from New England Biolabs (Ipswich, MA). The additional *Methylobacterium* strains presented here were either isolated from the University of Washington campus (L1, D21, D23, D24; CJM, unpublished) or from the surroundings of Woods Hole, MA (J-4-1, C-7-2, C-7-1, C-2-3, G-1-1, M-1-1; NF Delaney, unpublished). Strains and plasmids are listed in *Supplementary file 1*.

## Experimental evolution

Transconjugant strains containing pJM10 were streaked to single colonies on M-PIPES + succinate + kanamycin plates. Liquid cultures were inoculated from single colonies into M-PIPES + succinate + kanamycin and grown to saturation. Cultures were then diluted 100× into M-PIPES + DCM to initiate the evolution experiment, starting 3–4 replicate flasks for each transconjugant. Every 3.5 days, cultures were diluted into fresh M-PIPES + DCM. Initial dilution factors were 16×, rising to 64× by the end of the experiment. Aliquots were taken at 12, 24, 36, 48, 60, 90, 120, and 150 generations and frozen at −80°C in 8% DMSO. After 150 generations, the cultures were plated on M-PIPES + succinate plates to isolate single colonies. Individual colonies were restreaked onto M-PIPES + succinate plates, then inoculated into liquid cultures in M-PIPES + succinate. Liquid cultures were diluted 100× into M-PIPES + DCM to test for DCM growth. For each replicate population, the isolate with the highest yield on DCM as assayed by $OD_{600}$ was selected for further analysis.

The ancestral AM1 transconjugant was unable to grow on DCM alone. Consequently, the initial stages of evolution were performed in DCM supplemented with 0.75 mM methanol and kanamycin. For two of the populations, A2 and A3, the populations were able to grow on DCM alone after 60 generations of evolution, and the remaining 90 generations were conducted on DCM alone. For the final population, A1, the entire evolution experiment was performed with DCM + methanol + kanamycin.

## Competitive fitness assays

Competition experiments were conducted to compare the fitness of the tested strain against *M. extorquens* DM4 *ΔdcmA hptA::Venus* + pJM10, as described previously (*Michener et al., 2014*). Briefly, strains were grown to saturation in M-PIPES + succinate + kanamycin, and then diluted 100× into M-PIPES + DCM. After 3 days, cultures were mixed at defined ratios in fresh M-PIPES + DCM and allowed to grow for 3 more days. A flow cytometer was used to measure the population ratios before and after the final round of growth on DCM. The fitness of the isolate was then calculated based on the change in population ratios and the fold-growth of the mixed population.

## Genome resequencing

To prepare genomic DNA for sequencing, evolved isolates were streaked to single colonies on M-PIPES + succinate, and then grown to saturation in M-PIPES + succinate. Saturated cultures were centrifuged at 4500×*g* for 10 min, washed in 1 ml of water, and centrifuged at 8000×*g* for 3 min. After discarding the remaining supernatant, cell pellets were stored at −20°C overnight. Pellets were resuspended in 570 µl of TET (10 mM Tris pH 7.5, 1 mM EDTA, and 1% Triton X-100), heated to 90°C for 1 hr, then cooled to RT. Lysozyme was added to a final concentration of 2 mg/ml, and the suspension was incubated at 37°C for 30 min. Next, 30 µl of 10% SDS and 0.2 mg/ml of proteinase K were added, and the suspension was incubated for a further 1 hr at 37°C. After addition of 0.5 mg/ml RNAse A, the suspension was incubated at 37°C for 1 hr, heated to 90°C for 10 min, and 100 µl 5 M NaCl and 80 µl CTAB/NaCl (10% CTAB in 0.7 M NaCl) were added. After a further 10 min incubation at 90°C, the mixture was extracted twice with phenol/chloroform, precipitated with isopropanol, air dried, and resuspended in 10 mM Tris pH 7.5.

DNA sequencing was performed at the Microarray and Genomic Analysis Core Facility of the University of Utah and the IBEST Genomics Resources Core of the University of Idaho using a HiSeq 2000 and yielding 90×–200× coverage. Genomic DNA libraries were constructed using a TruSeq DNA Sample Prep LT kit (Illumina, San Diego, CA) following the manufacturer's instructions. Genome sequences were analyzed using breseq-0.24 (*Deatherage and Barrick, 2014*). Putative mutations were confirmed by PCR amplification from the chromosome and Sanger sequencing. To determine the first appearance of a given mutation, the chromosomal loci were amplified by PCR from the mixed population aliquots that had been frozen at intermediate time points during evolution. The fraction of the population with a given allele was determined by Sanger sequencing of these mixed PCR products, with an estimated detection limit of ~5% of the population.

## Plasmid and strain construction

To construct plasmid pJM83, plasmid pJM40 was amplified in an around-the-horn PCR to remove the *pHluorin/mCherry* coding frame. The *clcA* gene was amplified from PA1 gDNA with primers that added homology to the pJM40 expression construct (promoter and terminator). The *clcA* insert was then cloned into the pJM40 PCR product using Gibson assembly (*Gibson et al., 2009*).

To construct the allele exchange vectors, plasmid pPS04 was digested with *Xba*I and *Sac*I. The appropriate chromosomal locus, including ~500 bp on either side of the desired mutation, was amplified by PCR using primers that added homology to the pPS04 vector backbone. The chromosomal amplicon was then cloned into the pPS04 backbone using Gibson assembly (*Gibson et al., 2009*). In the case of plasmids pJM66, pJM67, pJM75, pJM76, pJM88, and pJM89, the mutant allele was moved into a strain in which it did not occur. Rather than cloning a single ~1000 bp chromosomal locus, these plasmids were constructed from three separate amplicons. These amplicons included a ~500 bp upstream homology region amplified from the recipient chromosome, a small region containing the desired mutation amplified from the donor chromosome, and a second ~500 bp downstream homology region amplified from the recipient chromosome. The PCR primers used added the appropriate homology to combine all three amplicons into the pPS04 backbone in a single Gibson assembly reaction.

Plasmids were mated into recipient *Methylobacterium* strains using triparental matings as described previously (*Fulton et al., 1984*). Allele exchanges were performed as described (*Marx, 2008*). Putative allele exchange mutants were confirmed by Sanger sequencing a PCR amplification of the modified chromosomal locus.

## In vivo pH measurements

In vivo pH measurements were conducted as described previously (*Michener et al., 2014*). Briefly, an expression plasmid containing both *dcmA* and a pHluorin-mCherry translational fusion (pJM40) was mated into the appropriate strain. The transconjugant was grown to saturation in M-PIPES + succinate + tetracycline, then diluted 100× into M-PIPES + DCM, and grown for three days. Cultures were diluted to a final optical density of 0.01 before analysis on an LSRII flow cytometer (BD, Franklin Lakes, NJ). pHluorin was excited at 488 nm and measured at 530/30 nm. mCherry was excited at 561 nm and measured at 620/40 nm. Samples were gated for forward scatter, side scatter, and mCherry fluorescence. After determining the population fluorescence, DCM was added to a final concentration of 10 mM. The culture was quickly vortexed and returned to the flow cytometer, leading to an approximately 20 s delay between DCM addition and consistent fluorescence measurements. After DCM addition, the population fluorescence was monitored for a further 10 min.

For each strain, a standard curve was constructed by diluting the DCM-grown culture to a final $OD_{600}$ of 0.01 in a solution composed of 30 mM buffer, 50 mM NaCl, 3 mM KCl, 10 µM valinomycin, and 10 µM nigericin. Buffers used were MES pH 5.1, MES pH 5.3, MES pH 5.5, MES pH 5.7, MES pH 6.1, PIPES pH 6.5, PIPES pH 6.9, and PIPES pH 7.3. The population mean fluorescence ratio (pHluorin fluorescence divided by mCherry fluorescence) was measured for each combination of strain and pH, then fit to a modified Henderson–Hasselbalch equation. The internal pH of the experimental samples was calculated by finding the fluorescence ratio of each cell, dividing the timecourse into 5 s intervals, calculating the population mean of the fluorescence ratio for each interval, and comparing that ratio to the standard curve to calculate the internal pH.

## qRT-PCR measurements

For qRT-PCR measurements, cultures were grown to saturation in M-PIPES plus succinate, and then diluted into the indicated media conditions. Upon reaching mid-log phase (typically at half the optical density of a saturated culture), cultures were centrifuged for 10 min at 4500×*g* and 4°C, washed once with 1 ml of water, and pelleted again at 8000×*g* for 3 min. Cell pellets were flash frozen in liquid nitrogen and stored at −80°C overnight.

RNA was extracted using an RNeasy Kit (Qiagen, Germantown, MD) and RNAse-free DNAse (Qiagen) according to the manufacturer's instructions. Total RNA was reverse transcribed using SuperScript III (Life Technologies, Carlsbad, CA), approximately 1 µg of RNA, and gene specific primers (rpsB FWD 5′-ACCAACTGGAAGACCATCTC-3′; rpsB REV 5′-CTTCTCGAGCTTGTCCTTCTCAC-3′; clcA FWD 5′-ATCGTCACCGAGATGACCCAG-3′; clcA REV 5′-CCAAGGTGTGATAGAGGCCG-3′) according to the manufacturer's directions. cDNA was quantified by qPCR, using a CFX-96 qPCR machine (Bio-Rad, Hercules, CA) and EvaGreen qPCR mix (Biotium, Hayward, CA). Three technical

replicates were performed for each biological replicate. For each replicate, the *clcA* concentration was normalized by the *rpsB* concentration. For each strain, the average normalized *clcA* concentration was compared to the reference strain, typically PA1.

## Growth rate measurements

Growth rates were measured using an automated system as described previously (*Delaney et al., 2013b*). Briefly, strains were grown to saturation in M-PIPES + succinate and then diluted 64× into 640 µl of M-PIPES + succinate with the appropriate concentration of osmolyte (NaCl or sucrose) in a 48-well plate. After the cultures reached saturation, they were again diluted 64× into 640 µl of the appropriate media, with three replicate wells per condition. The optical density of these assay plates was monitored every 45 min until the cultures again reached saturation. The growth rate of the culture was calculated using CurveFitter (*Delaney et al., 2013a*).

## Acknowledgements

The authors acknowledge financial support from the National Institutes of Health (F32 GM106629 to JKM), the Fundação de Amparo à Pesquisa do Estado de São Paulo, and the Coordenação de Aperfeiçoamento de Pessoal de Nível Superior (fellowships to AACN). The authors thank Y Trotsenko for providing *M. extorquens* DM17, C Miller for advice on testing ClcA directionality, DD Nayak for outlining several of the early concepts of this project, NF Delaney for providing six *Methylobacterium* isolates, and members of the Marx Laboratory for offering helpful comments on the manuscript. Genome sequencing at the University of Idaho IBEST Genomics Resources Core was supported by grants from the National Center for Research Resources (P20RR016448) and the National Institute of General Medical Sciences (P20GM103397) from the National Institutes of Health.

## Additional information

### Funding

| Funder | Grant reference number | Author |
|---|---|---|
| National Institutes of Health | F32 GM106629 | Joshua K Michener |
| São Paulo Research Foundation | Fundacao de Amparo a Pesquisa do Estado de Sao Paolo | Aline A Camargo Neves |
| Coordenação de Aperfeiçoamento de Pessoal de Nível Superior | | Aline A Camargo Neves |

The funders had no role in study design, data collection and interpretation, or the decision to submit the work for publication.

### Author contributions

JKM, Conception and design, Acquisition of data, Analysis and interpretation of data, Drafting or revising the article; AACN, Acquisition of data, Analysis and interpretation of data, Drafting or revising the article; SV, FB, CJM, Conception and design, Analysis and interpretation of data, Drafting or revising the article

## Additional files

### Supplementary file

• Supplementary file 1. Strains and plasmids used in this study.

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
