## [Decision Letter]

Thank you for sending your work entitled “Effective use of a horizontally-transferred pathway for dichloromethane catabolism requires post-transfer refinement” for consideration at *eLife*. Your article has been favorably evaluated by Detlef Weigel (Senior editor), Michael Laub (Reviewing editor), and 2 reviewers.

The Reviewing editor and the reviewers discussed their comments before we reached this decision, and the Reviewing editor has assembled the following comments to help you prepare a revised submission.

We all generally liked this manuscript and felt that it reported an interesting story with conclusions well-justified by the data shown. There were a few issues with regard to the writing that need attention in a revised manuscript. There were also a couple of issues raised in the reviews and during our subsequent discussion that need to be addressed experimentally. These primary issues are summarized here and, with one possible exception, are straightforward experiments to add:

1) We each had questions about the nature of the naturally occurring mutations in clcA in DM4 and DM17. You should provide stronger evidence that these mutations have the anticipated effects/consequences on clcA expression levels.

2) What are the fitness costs of dcmA and refinement alleles under non-DCM selective conditions?

3) Can you include FreqSeq data or at least some sampling of intermediates from the evolved lines to probe temporal dynamics of accumulated mutations?

4) One final item that came up is whether the clcA mutations identified actually make the transporter leaky, as predicted, leading to better chloride ion export. It was not clear to us how involved this type of experiment would be, but if it's reasonably straightforward it would be good to add this as well. You reference a chloride ion transport study in your paper and perhaps the approach taken previously could be implemented to solidify your conclusions about the clcA mutants.

---

## [Author Response]

*1) We each had questions about the nature of the naturally occurring mutations in clcA in DM4 and DM17. You should provide stronger evidence that these mutations have the anticipated effects/consequences on clcA expression levels*.

We have revised Figure 5 to include new qRT-PCR data showing the clcA expression in DM17 and in the promoter exchange strains. We can now show that both of the natural isolates, DM4 and DM17, have high clcA expression. Additionally, the promoter exchange mutants have the expected trend in clcA expression.

2) What are the fitness costs of dcmA and refinement alleles under non-DCM selective conditions?

We have added a new figure (Figure 3—figure supplement 2) and explanatory sentence describing the fitness effects of these mutations during growth on succinate. In short, none of the mutations have major fitness effects during growth on succinate compared to large effects on DCM.

3) Can you include FreqSeq data or at least some sampling of intermediates from the evolved lines to probe temporal dynamics of accumulated mutations?

We have added two new supplementary figures, Figure 3—figure supplement 3 and Figure 3—figure supplement 4, to show the allele frequency dynamics of selected populations. Of note, we can connect changes in allele frequency of the *M. nodulans* isolates to changes in bulk properties of the evolving population, such as the yield on DCM. Additionally, we include allele frequency measurements for the single isolate, E2, in which we find two highly beneficial mutations (as shown in Figure 3). The remaining populations have less informative allele frequency dynamics, as we only observe a single mutation rapidly sweeping to fixation.

*4) One final item that came up is whether the clcA mutations identified actually make the transporter leaky, as predicted, leading to better chloride ion export. It was not clear to us how involved this type of experiment would be, but if it's reasonably straightforward it would be good to add this as well. You reference a chloride ion transport study in your paper and perhaps the approach taken previously could be implemented to solidify your conclusions about the clcA mutants*.

We assume that the reviewers were referencing the mutations to secY that we propose yield a leaky anion channel. While we agree that directly measuring the chloride permeability of inverted membrane vesicles containing the various SecY mutants would strengthen our hypothesis, running the necessary experiments would be quite challenging. We estimate that it would take an additional collaborator and a year’s work from a PhD student to run these experiments as there is no precedent for doing any of this work with inverted vesicles in Methylobacterium, or indeed for many organisms outside of *E. coli*. Instead, we direct the reviewers’ attention to our in vivo experiments where we demonstrate increased sensitivity to external NaCl as an alternative demonstration of facilitated chloride diffusion.